# Extinction cascades, community collapse, and recovery across a Mesozoic hyperthermal event

Alexander M. Dunhill [1] ✉, Karolina Zarzyczny [1,2,3,4], Jack O. Shaw[5,6], Jed W. Atkinson[1,7], Crispin T. S. Little [1,4] & Andrew P. Beckerman [8]

Mass extinctions are considered to be quintessential examples of Court Jester drivers of macroevolution, whereby abiotic pressures drive a suite of extinctions leading to huge ecosystem changes across geological timescales. Most research on mass extinctions ignores species interactions and community structure, limiting inference about which and why species go extinct, and how Red Queen processes that link speciation to extinction rates affect the subsequent recovery of biodiversity, structure and function. Here, we apply network reconstruction, secondary extinction modelling and community structure analysis to the Early Toarcian (Lower Jurassic; 183 Ma) Extinction Event and recovery. We find that primary extinctions targeted towards infaunal guilds, which caused secondary extinction cascades to higher trophic levels, reproduce the empirical post-extinction community most accurately. We find that the extinction event caused a switch from a diverse community with high levels of functional redundancy to a less diverse, more densely connected community of generalists. Recovery was characterised by a return to pre-extinction levels of some elements of community structure and function prior to the recovery of biodiversity. Full ecosystem recovery took ~7 million years at which point we see evidence of dramatically increased vertical structure linked to the Mesozoic Marine Revolution and modern marine ecosystem structure.

Earth has experienced a number of mass extinction events that have shaped the evolutionary history of biodiversity and ecosystem function by the dramatic loss of species over relatively short periods of time and by the associated restructuring of ecosystems[1]. Many mass extinctions are linked to Large Igneous Province (LIP) volcanism which is defined by rapid global warming, ocean anoxia, and ocean acidification[2]. Palaeobiologists have long viewed these events and drivers of mass extinctions as the "Court Jesters" of macroevolution and biodiversity across long geological timescales[3]. Macroecological studies of selectivity[4,5] and functional diversity loss[6,7] support this. They indicate that warming-related mass extinctions were biased by latitude[4,8–10] and that taxa vulnerable to hypercapnia, anoxia, and acidification[4,9,11,12] were most strongly affected.

However, a large portion of this work on extinctions has ignored the role of species interactions[13]. Species interactions are central to understanding who, when and why species go extinct because they can buffer or accentuate extinction risk. They are also central to how "Red Queen" processes that link speciation rates to extinction rates[14] may

[1]School of Earth and Environment, University of Leeds, Leeds, UK. [2]School of Biology, University of Leeds, Leeds, UK. [3]School of Ocean and Earth Science, National Oceanography Centre, University of Southampton, Southampton, UK. [4]Department of Life Sciences, Natural History Museum, London, UK. [5]Department of Earth and Planetary Sciences, Yale University, New Haven, CT, USA. [6]Santa Fe Institute, Santa Fe, NM, USA. [7]Leeds Museums and Galleries, Leeds, UK. [8]School of Biosciences, Ecology and Evolutionary Biology, University of Sheffield, Sheffield, UK. ✉e-mail: a.dunhill@leeds.ac.uk

underpin our understanding of the recovery of biodiversity, structure and function after mass extinction events[15–18]. Thus, modern ecological theory and the importance of extinction in the Red Queen hypothesis suggest that extinction and recovery dynamics to such hyperthermal events are likely to be most effectively evaluated via a community ecological framework[18].

With respect to extinction, many victims of mass extinctions were unlikely to have become extinct as a direct effect of abiotic stress, but probably did so in response to cascading secondary effects[19,20]. If we are to truly understand mass extinction dynamics, including the high levels of extinction amongst pelagic predators[4] which are difficult to explain in the absence of extinction cascades through communities[21], we must embed interaction networks in our analyses of extinction patterns.

Embedding interaction networks in the analysis of ecosystem recovery is also warranted. Studies of recovery from these events based solely on taxonomic and functional diversity suggest full ecosystem recovery can take anywhere between <1 to 50 million years from the largest mass extinctions[22–24]. Yet it is possible that ecosystem function could recover despite persistent low levels of biodiversity. Furthermore, taxonomic approaches do not resolve changes in community structure, which are tied to key palaeoecological hypotheses about the transition to modern ecological community structure, and require detail on the diversity, structure and functioning of communities.

In this study, we focus on the Early Toarcian Extinction Event (ETEE; ~183 Ma), a second order extinction event[25] (i.e., an extinction event that caused less than 40% generic extinction globally[26]) that resulted in the loss of around 26% of marine genera globally[27]. It was linked to the eruption of the Karoo–Ferrar Large Igneous Province[28], which likely resulted in a globally distributed negative carbon isotope shift[29,30], hyperthermal warming of up to 13 °C in the mid-latitudes[31,32], prolonged regional ocean dysoxia and anoxia[27,33–35], and ocean acidification[36]. In the Cleveland Basin, the ETEE is coincident with the deposition of finely laminated, organic-rich, black shales which signify persistent dysoxia/anoxia at shallow depths on the continental shelf (i.e., the Toarcian Oceanic Anoxic Event[34,37]). The ETEE resulted in the loss of around 60% of marine species within the Cleveland Basin (87% benthic species extinction)[25,38].

Here, we formally embed community structure and details about species interactions into a palaeoecological analysis of the ETEE from the Cleveland Basin, Yorkshire, UK to resolve detail about which, when and why species went extinct and the transition of community recovery towards modern marine ecosystem structure. We use a data set of 38,670 occurrences of 162 species of marine invertebrates (ammonites, belemnites, bivalves, brachiopods, decapod crustaceans and echinoderms), fish, and trace fossils derived from years of detailed field studies (see Methods) of one of the most expanded Pliensbachian to Toarcian sections in the world to produce a series of community trophic networks (i.e., food webs) (Fig. 1). Specifically, we use ecological trait data to reconstruct plausible food webs. We then subject these food webs to several primary extinction scenarios that link event characteristics (e.g., dysoxia, acidification, warming) to traits. We use well-established ecological modelling tools to evaluate scenarios of primary and patterns of secondary extinction, ultimately identifying several target traits and species whose sensitivity to the ETEE event likely led to the ensuing post-event community structure. Finally, we also look at empirical patterns of recovery from this extinction event, detailing how functional groups, motifs of species interaction, community structure and stability (measured as robustness to extinction) recovered at different rates to diversity.

## Results and discussion
We combined a trait-based food web reconstruction method with modern secondary extinction modelling (see Methods) to estimate the most plausible targets (traits and taxa) of primary extinction that drove loss of community diversity and structure across the ETEE via a mix of primary and secondary extinction. Secondary extinctions are defined in our analyses by when a consumer loses all its prey items. The secondary extinction modelling generates 13 scenarios arising from targeting six traits in two orders (e.g., small to big, and big to small for body size) and a random extinction scenario (see Methods). We first document 'who died and why', comparing across the ETEE the predicted loss of guilds from our secondary extinction modelling to existing empirical knowledge of guild loss over this time period. We also focus on predicted and actual changes in 13 community structure metrics across the ETEE event. We then assess the actual changes in community structure and function alongside analysis of food web robustness as a proxy for stability, across all four reconstructed networks from pre-extinction to full recovery phase.

### Who died and why?
We found that primary extinction selectivity based on tiering (i.e., where in the water/sediment column an organism resides), with strongest extinction selectivity against infaunal (i.e., organisms living within the sediment) versus pelagic taxa, gave by far the closest replication of the empirical post-extinction community (see Methods for definition of community in this study). High levels of primary extinction selectivity against infaunal and epifaunal taxa reflect the dysoxic environment in the immediate post-extinction interval, which would be much more intensely felt by organisms living in or on the seabed sediments. The strength of our inference arises from the use of a True Skills Statistic (TSS) (Fig. 2a) where high values represent a match in diversity (guild richness) and the identity of taxa lost. Our inference is further strengthened by comparing empirical and simulated post-extinction community structure across 13 structural food web metrics tied to community complexity, generalism/specialism and interaction motifs defining the relative abundance of competition and vertical trophic interactions. This multi-metric comparison also reveals that primary extinction selectivity against tiering (infaunal > pelagic) produces, on average, the closest match to the empirical post-extinction community (Fig. 2b). Secondary extinctions were primarily concentrated amongst secondary consumers in the benthic realm (e.g., crustaceans, echinoderms etc.) as their primary consumer prey were drastically reduced due to primary extinction selectivity (Fig. S1).

The next two closest scenarios in terms of replicating empirical post-extinction community structure by TSS and mean difference across 13 structural metrics are linked to generalism of the consumers. Both primary extinction scenarios selecting against the most generalist consumers (H to L) and the least generalist consumers (L to H) perform equally well and produce matches to the empirical post-extinction community that are less accurate than the tiering (infaunal>pelagic) scenario but better than random primary extinction selectivity (Fig. 2a, b). Primary extinction selection from high to low generalism was driven by the loss of benthic intermediate consumers (i.e., crustaceans, gastropods, echinoderms) which fed across multiple trophic levels and were thus at risk from secondary extinction cascades as well as their active predatory life habits being hard to maintain under low oxygen conditions[39,40]. These scenarios also generated secondary extinction cascades up to higher trophic levels and the loss of tertiary benthic and pelagic predators (Fig. S1). Primary extinction selection from low to high generalism is reflective of specialist sensitivity where, in this case suspension feeders, were primarily epi- and infaunal seabed dwellers which were most at risk from the dysoxic conditions. This scenario generated secondary extinction cascades that are similar to the tiering scenario, where primary selection centred on suspension feeders caused secondary extinction of benthic secondary consumers (Fig. S1). Other traits, including some that have previously been associated with being key determinants of extinction across hyperthermals

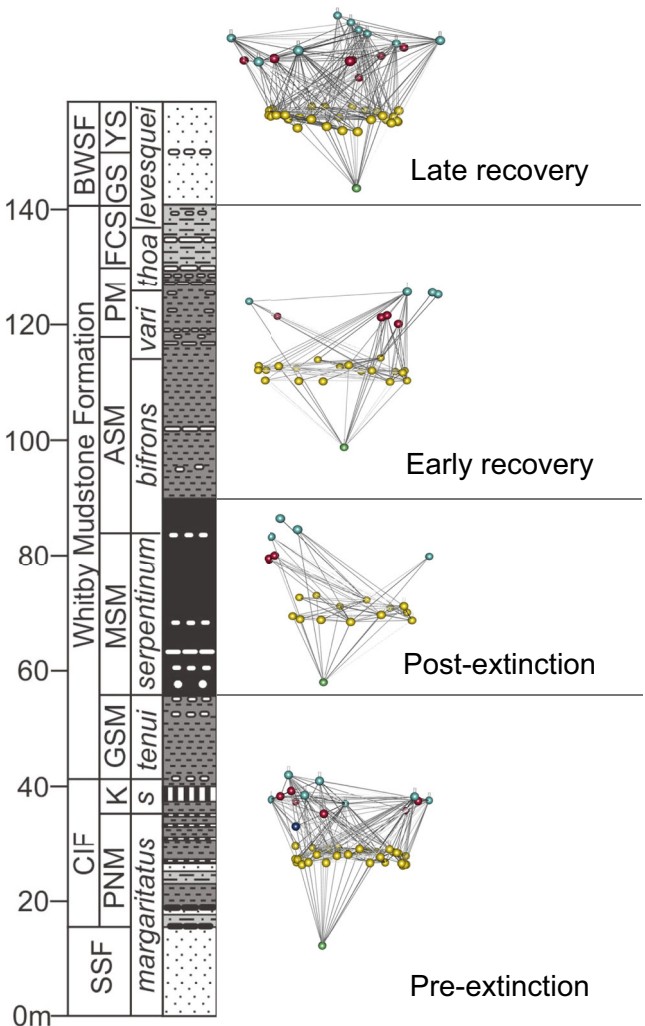

**Fig. 1 | Stratigraphic column of the Pliensbachian–Toarcian (Lower Jurassic) of the Cleveland Basin at Ravenscar (North Yorkshire, UK) showing community food webs for pre-extinction, post-extinction, early recovery and late recovery intervals.** Node colours represent broad trophic levels. SSF Staithes Sandstone Formation, CIF Cleveland Ironstone Formation, BWSF Blea Wyke Sandstone Formation, PNM Penny Nab Member, K Kettleness Member, GSM Grey Shales Member, MSM Mulgrave Shales Member, ASM Alum Shales Member, PM = Peak Mudstone Member, FCS Fox Cliff Siltstone Member, GS Grey Sandstone Member, YS = Yellow Sandstone Member, *s spinatum, vari variabilis, thoa thouarsense, tenui tenuicostatum.*

generalists, lower frequencies of apparent and exploitative competition, increased vertical structure (e.g., more linear food chains) and more omnivory (Fig. 2c, d). These results are comparable to patterns seen across other extinction events and incidences of biodiversity loss in both the fossil record[43,44] and more recent ecological record[20,45] whereby the loss of function guilds resulted in reduced structural complexity[43]. The pre-extinction community was a diverse assemblage of benthic and pelagic taxa[41,46,47] and was characterised by values of many common network metrics that are well within the bounds for typical modern day marine communities including connectance, generality, vulnerability, max. trophic level etc.[19,48] (Fig. 2c, d).

Previous work[41] suggests that infaunal benthic species experienced large losses. Our evidence above corroborates these details as simulations that target infaunal benthic species generated the best match between predicted and actual community data (Fig. 2a, b). The empirical data[41] also indicates that large, highly motile, and predatory benthic guilds (i.e., crustaceans, echinoderms etc.) also went extinct (~80% benthic species extinction)[41]. Our simulations suggest that secondary extinctions tied to the infaunal or generalism primary extinction targets (Fig. S1) as well as the dysoxia, are a highly plausible mechanism driving mass extinction outcomes. Dysoxia is well evidenced over this time period in the Cleveland Basin[25,41], and would make an active, predatory benthic lifestyle difficult to maintain[40].

This led to a post-extinction benthic assemblage dominated by low-diversity/high-abundance communities of small, epifaunal, suspension-feeding bivalves, most notably the presumably low-oxygen-tolerant opportunistic species *Bositra buchii* and *Pseudomytiloides dubius*[35,41]. This post extinction community bears a broadly similar structure to that of modern low-diversity communities[48]. The network is characterised by densely connected, species-poor communities of opportunists/generalists which is also consistent with evidence from palaeoecological interpretations of the fossil record (i.e., low-diversity/high-abundance communities of opportunistic species)[35,38,41] and other unstable post-mass extinction food webs reconstructed from the fossil record[16,17,49].

The details of this are captured by a number of key metrics linked to the number and distribution of links among species in the community: overall community connectivity (i.e., connectance) increased after the ETEE (Fig. 2c) which corresponds with an increase in generality (average number of prey), vulnerability (average number of predators), and maximum trophic level (Fig. 2c). Together with an increase in the number of linear chains within the food web (S1), in the levels of omnivory (S2) and reductions in both apparent (S4) and direct competition (S5) (Fig. 2d), this suggests that the post-extinction community showed much reduced functional redundancy (i.e., the Skeleton Crew hypothesis)[6,50] with taxa post ETEE being more generalist in their feeding habits and thus more closely linked to one another via consumer-resource interactions than taxa in the pre-extinction community.

Selective extinction of benthic taxa, which are predominantly lower- and intermediate-level consumers, led to the food web becoming more dominated by linear species interactions (i.e., more linear chains; S1), fewer competitive motifs (i.e., fewer S4 and S5) and increased omnivory (i.e., more S2). Collectively, this increased the prevalence of vertical indirect effects, e.g., the potential for top consumers to affect non-adjacent trophic levels (Fig. 2d). We suggest that reduced direct competition (S5) arose as benthic predators disappeared from the community and reduced apparent competition (i.e., predator choice; S4) arose as the extinction wiped out the majority of the benthic guilds such that predators had fewer prey options. The higher levels of omnivory likely arose as the intermediate consumers in the benthic realm were lost, meaning that the few remaining predators were feeding across more trophic levels to obtain enough food (Fig. 2d).

(i.e., body size and calcification), did not produce simulated post-extinction communities that were a closer match to the empirical post-extinction communities than random selection (Fig. 2a, b).

Our overall result aligns with empirical evidence that the high levels of extinction occurred within the benthic realm[25,27,41,42] and also conforms to the general consensus that the ETEE in the Cleveland Basin was primarily driven by an anoxia/dysoxia kill mechanism. Here we detail further unique insights that arise from our inference based on the 13 structural metrics.

## Structural and functional changes across the ETEE

The transition across the ETEE is characterised by a switch from a diverse, stable community with high levels of functional redundancy (i.e., multiple guilds occupying broad life habits e.g., benthic suspension feeders) to a smaller food web (lower richness) with a more dense network of interactions (complexity) characterised by more

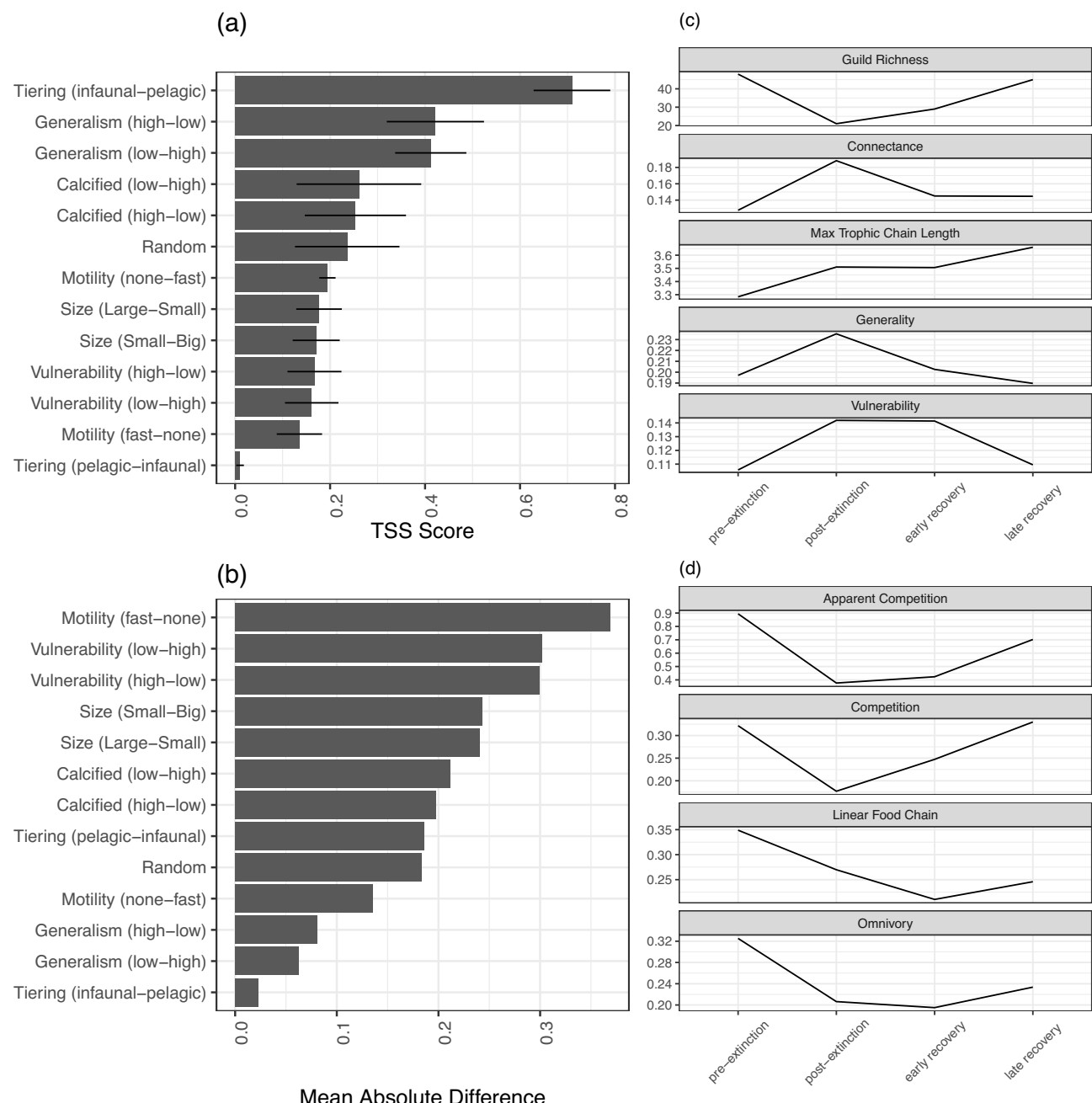

**Fig. 2 | Comparison of the structure of modelled and empirical post-extinction food webs and empirical food web structure and motifs throughout the pre-extinction, post-extinction, early recovery and late recovery intervals. a** TSS scores showing the match of diversity and identity of taxa lost between modelled extinction scenarios and the empirical post-extinction community. A score of closer to 1 represents a better match. **b** Mean difference across 13 food web metrics between modelled extinction scenarios and the empirical post-extinction community. **c** Structural food web metrics in the empirical food webs across the extinction and recovery interval. **d** Functional interaction motifs in the empirical food webs across the extinction and recovery interval.

## Structure function and diversity changes through the recovery interval

These same 13 metrics allow for analyses of whether biodiversity, structure and function recovered in tandem or not after the extinction event. These metrics allow us to assess whether there are any substantive changes in community structure leading to signatures of the Marine Mesozoic Revolution and the emergence of modern marine ecosystem structure[51].

The recovery interval from the ETEE features two distinct patterns. The first is that guild richness, connectance (complexity), generality, vulnerability and competitive motif structure returned to pre-

extinction event states, but this takes ~7 million years (Fig. 2c, d). This mirrors data on post-extinction species diversity recovery[41] and this length of time is close to estimates for much larger and more global extinction events such as the Permo-Triassic mass extinction[22]. This suggests that the ETE was a regionally intense event in the Cleveland Basin, even if not consistently so across the entire globe[52].

The second pattern is that there was a noticeable increase in maximum trophic level and the associated motifs of food chains and omnivory, representing much more vertical structure in the late recovery community relative to the pre-extinction community (Fig. 2c, d). This novel trophic structure represented via vertical motifs

and maximum trophic level in the late recovery interval suggests that marine communities had attained a new stable state due to ongoing macroevolutionary changes, i.e., the Mesozoic Marine Revolution (MMR)[53]. In the move towards the Middle Jurassic, newly diversifying plankton groups[54], greater diversity of benthic predators and the emergence of giant marine apex predators suggests increased nutrient fluxes supporting taller and more complex marine food webs[51,55].

Aspects of community diversity, structure and function had variable recovery times after the ETEE in the Cleveland Basin. Connectance and generality (Fig. 2c) are the two metrics that returned most quickly to pre-extinction levels by the early recovery interval. In contrast, guild richness did not return to pre-extinction levels until the late recovery interval and a similar pattern of long recovery is seen in structural metrics such as vulnerability (Fig. 2c) and the functional motifs of apparent and direct competition (Fig. 2d). The early recovery interval saw a slight increase in richness as a number of guilds returned that were absent from the basin during the post-extinction interval following re-oxygenation of the benthic realm (Fig. 2c)[41]. However, despite the return of some species occupying motile benthic and infaunal guilds, the majority of new species occupied guilds that were also present during the immediate post-extinction interval (i.e., surficial suspension feeders and pelagic predators)[41].

The relationship between connectance and species richness and connectance and other structural metrics in modern extant communities have some well-known patterns[56]. First, connectance goes down with increased richness, so it is unsurprising that connectance increased across the ETEE and then recovered back towards pre-extinction levels through the recovery interval as guild richness increased (Fig. 2c). Second, generality is negatively correlated with connectance while maximum trophic level is positively correlated[56]. The pattern across the ETEE contradicts both of these relationships found in modern ecological systems. Instead, we see increased generality in line with increased connectance across the ETEE and increasing trophic height even when connectance declines (Fig. 2c).

Despite not conforming to what might be expected of modern ecosystem structure, an increase in generality across the ETEE is in line with mass extinction theory which suggests generalist taxa are more likely to survive mass extinction events and thus populate early recovery communities as specialists are more likely to be selected against[57,58] and are more vulnerable to secondary extinction cascades[15]. Furthermore, the increase in trophic height even as connectance declines (Fig. 2c) may arise because the late recovery stages experienced major macroevolutionary changes as the Marine Mesozoic Revolution delivered increases in productivity[51,55,59] and the evolution of new predators[51,55], two things supporting increased trophic height.

Whilst some structural metrics returned towards pre-extinction levels, full ecosystem recovery does not appear to have happened by the end of the early recovery interval (Fig. 2c, d). This is evidenced by a paucity of infaunal tiering and motile benthos, as compared to the pre-extinction interval[41], and several of the structural metrics and motifs remained at similar levels to the post-extinction interval rather than starting to return to pre-extinction levels (Fig. 2c, d). For example, maximum trophic level and vulnerability (Fig. 2c) remained very high with low levels of competition (Fig. 2d). This suggests that the early recovery community was tall, thin and top-heavy and consisted of a diverse assemblage of pelagic predators feeding on a still relatively depauperate assemblage of lower-level consumers. This pattern contrasts with previous models of ecosystem recovery following mass extinctions that postulate that lower trophic levels recovered prior to higher trophic levels[23]. Instead, this pattern supports the hypothesis of delayed benthic ecosystem recovery and top-heavy trophic pyramids following mass extinctions[22].

The late recovery interval witnessed guild richness and almost all the structural metrics and motifs return, or start to return, to levels seen in the pre-extinction community (Fig. 2c, d). Although many of the taxa are different (at species level) to those of pre-extinction community, most pre-extinction guilds are re-occupied by the late recovery interval[41]. Vulnerability has now joined connectance and generality at levels comparable to the pre-extinction community (Fig. 2c), as are levels of apparent (S4) and direct (S5) competition (Fig. 2d), which matches modern ecological theory that states levels of direct and exploitative competition increase with increased biodiversity[60]. This suggests that intraguild diversity and functional redundancy are recovering – the reconstructed network indicates a greater number of predators were feeding upon a greater number of prey species and thus increased competition for prey and predator choice simultaneously.

The appearance of a greater diversity of lower and intermediate-level consumers in the benthic realm in the late recovery interval drove an increase in the number of linear chains and omnivory, rather than a return towards the lower levels of the pre-extinction community (Fig. 2d). Together with a further rise in maximum trophic level, these changes in the late recovery phase suggest that the food web has much greater vertical complexity than the pre-extinction community (Fig. 2c, d). The increase in the number of linear chains in the late recovery as compared to the early recovery suggests that food web shape started to return to pre-extinction levels before a further change in ecosystem structure brought about by the progression of the MMR[51,53]. The MMR is widely regarded as an 'escalation event' where the evolution of new predatory guilds in the Jurassic-Cretaceous drove a predator-prey arms race that lead to the restructuring of marine communities away from tiered communities of sessile suspension feeders to modern marine communities of motile and infaunal guilds[53]. The late recovery interval contained a much more diverse array of both predatory and motile and infaunal benthic guilds from groups that were supposedly key drivers of the MMR, such as decapod crustaceans[61], gastropods[62] and echinoderms[63] and such changes in community composition may have driven some of the stepwise increase in maximum trophic level through the system, which deviates from the common pattern of perturbation before return to pre-extinction levels as seen in most of the other structural metrics and motifs (Fig. 2c, d).

## Stability across the ETEE and through recovery

The change in species richness and associated community structure metrics such as connectance, motifs, levels of generalism and omnivory are likely to have affected the stability and robustness of the communities[20]. Ecological theory linking richness and these metrics to stability, however, requires additional information on the average and distribution of interaction strengths[64,65]. Thus, in order to generate a formal assessment of the stability of each network, we implemented a generalised robustness analysis. Here we estimated, for each network, the proportion of the community that remains after a given proportion of random primary extinctions, where the final proportion is a product of primary and secondary extinctions. The most common version of this analysis estimates $R_{50}$, the proportion of species that need to be made extinct via primary extinction that leads to a 50% loss of all species in the network/food web as a result of primary, and secondary extinctions (see refs. 66,67). Here we estimate $R_x$ under replicated, randomised primary extinction sequences, where x ranges from 1–99% following Jonsson et al.[66] who proposed a gradient based approach to manage the possibility that the threshold of choice (e.g., $R_{50}$) might hide variation in robustness that arises from the sensitivity of networks to the deletion sequences chosen.

Using this method, all networks experienced secondary extinctions because of primary extinctions (Fig. 3). More specifically, in all four networks a small percentage of primary extinctions can lead to 40–50% loss of species. We highlight two additional patterns. First, the transition across the ETEE (Pre- to Post-, Fig. 3) is not marked by a substantial change in the pattern of robustness suggesting that the

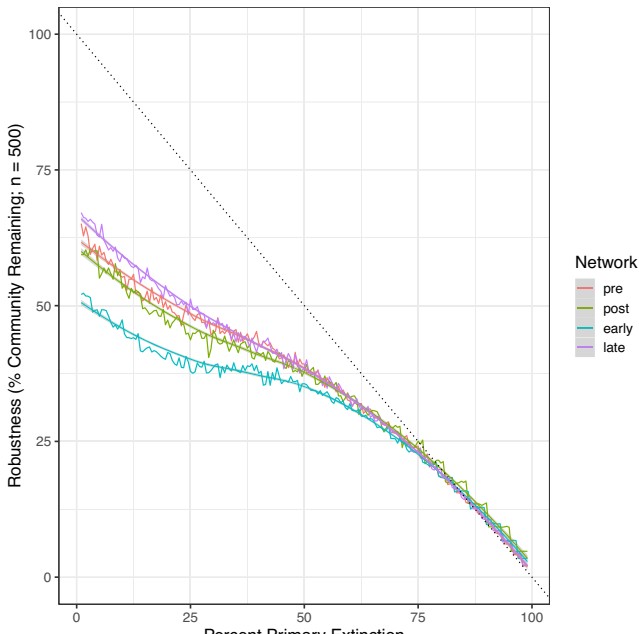

**Fig. 3 | Robustness (defined here as the % of the community remaining after a proportion of primary extinctions) varies with primary extinction proportion and among the networks.** The x-axis represents the proportion of the community removed via random primary extinctions. The y -axis is the proportion of the original community remaining. 1-y is the proportion lost due to primary and secondary extinction. For each value of primary extinction (x), we estimated robustness for 500 random primary deletion sequences, report the mean value and fit a loess smoother to visualise the patterns.

extinctions that did occur across the ETEE did not alter the community structure in such a way as to greatly modify overall robustness. Second, the transition from post ETEE to early recovery is accompanied by a further, more substantial reduction in robustness and this is then followed by a marked increase in robustness in the late recovery stage that is greater than the robustness of the pre-extinction community. This suggests that the recovery stage is marked by speciation and turnover of species (macroecological and evolutionary processes) that did alter the structure-robustness relationship. It is worth noting at this point, as detailed in our Methods, that all of our simulations, including this $R_x$ estimation, is based around a 'feasible' characterisation of the network where a single food web for each time interval consisting of all feasible feeding interactions is created as determined by the functional traits of the organisms and the feeding rules. Further work with modern and paleo- data will benefit from considering the assumptions, consequences and benefits of available network reconstruction methods that span the feasible - realised (mechanistic) spectrum (e.g., refs. [68–70]).

Furthermore, we hypothesise that this 'recovery' to increased levels of robustness is tied to the MMR and that it is plausible that modern marine ecosystems are more robust to secondary extinction cascades than more ancient marine ecosystems of the Palaeozoic and early Mesozoic. This may represent a new hypothesis as to why there has not been a mass extinction event (apart from the extra-terrestrial driven Cretaceous-Paleogene mass extinction) since the ETEE.

### Concluding remarks
Our comparison of multiple, trait-based primary extinction scenarios suggests that the ETEE in the Cleveland Basin was characterised by primary and secondary extinctions linked specifically to infaunal and epifaunal benthic guilds[41] and driven by shallow-water dysoxia/anoxia. These conclusions agree with lithological and geochemical evidence

for an anoxia/dysoxia kill mechanism[34,37,41,71] which would primarily target benthic organisms. There is also evidence that organisms with predatory life habits as well as specialist taxa were more at risk of going extinct because of metabolically demanding life habits that were hard to maintain under low oxygen levels as well as vulnerabilities to secondary extinction cascades.

The extinction event is further characterised by a switch from a diverse community where each key ecological function was performed by a number of guilds to a less diverse, more densely connected community of generalist "disaster taxa". This change from a diverse pre-extinction ecosystem with high degrees of functional redundancy to a contrasting post-extinction community where key functions are performed by single guilds is representative of the "Skeleton Crew Hypothesis"[6,50] in which the subsequent loss of any "crew member" may cause the system to collapse. Despite these characteristics, robustness of these two communities was not dramatically different.

The recovery interval from the ETEE was long and was represented by two distinct patterns; (1) diversity and ecosystem structure/function took up to 7 million years in all to return to levels seen in the pre-extinction community; (2) some ecosystem metrics suggest that marine communities attained a new state in the late Toarcian as a result of the Mesozoic Marine Revolution; and (3), robustness declined and then increased more substantially across this transition than during the ETEE itself. The late recovery interval was characterised by increased average trophic level, vertical complexity and robustness suggesting that micro- and macroevolutionary processes associated with the Mesozoic Marine Revolution, i.e., increases in primary productivity and greater predation pressure[51,53,54,72,73], had started to alter the structure of marine ecosystems by the end of the Early Jurassic.

## Methods
### Dataset
Fossil occurrence data was obtained from a compilation of field data sets[25,27,41,74–76]. The study interval extends from the upper Pliensbachian (-185 Ma) to the upper Toarcian (-175 Ma) of the Cleveland Basin (North Yorkshire, UK; Fig. 1) and provides a high-resolution data set across the ETEE. The data set consists of 38,670 specimens of 162 pelagic and benthic macroinvertebrate species together with occurrences of fish and trace fossils. The data set was subset into four broad time periods, or assemblages, which are treated as communities of interacting organisms; pre-extinction (*margaritatus-tenuicostatum* zones of the Staithes Sandstone Formation, Penny Nab and Kettleness Members of the Cleveland Ironstone Formation and majority of the Grey Shales Member of the Whitby Mudstone Formation), post-extinction (*serpentinum-commune* subzones of the Mulgrave Shale and Alum Shale Members of the Whitby Mudstone Formation), early recovery (upper *bifrons*-lower *levesquei* zones of the Alum Shale, Peak Mudstone and Fox Cliff Siltstone Members of the Whitby Mudstone Formation), and late recovery (upper *levesquei* zone of the Grey and Yellow Sandstone Members of the Blea Wyke Sandstone Formation) (Fig. 1).

### Defining organism ecologies, feeding interactions and trophic guilds
Modes of life were defined for each fossil species based on the ecological traits defined in the Bambach ecospace model[77] (i.e., motility, tiering, and feeding). Ecological traits were assigned based on interpretations from the published literature which are largely based on functional morphology and information from extant relatives. Information on the body size of each species was also recorded by summarising mean specimen sizes from the section into a categorical classification. The following ecological characteristics were recorded for each fossil species; motility (fast, slow, facultative, non-motile), tiering (pelagic, erect, surficial, semi-infaunal, shallow infaunal, deep infaunal), feeding (predator, suspension feeder, deposit feeder,

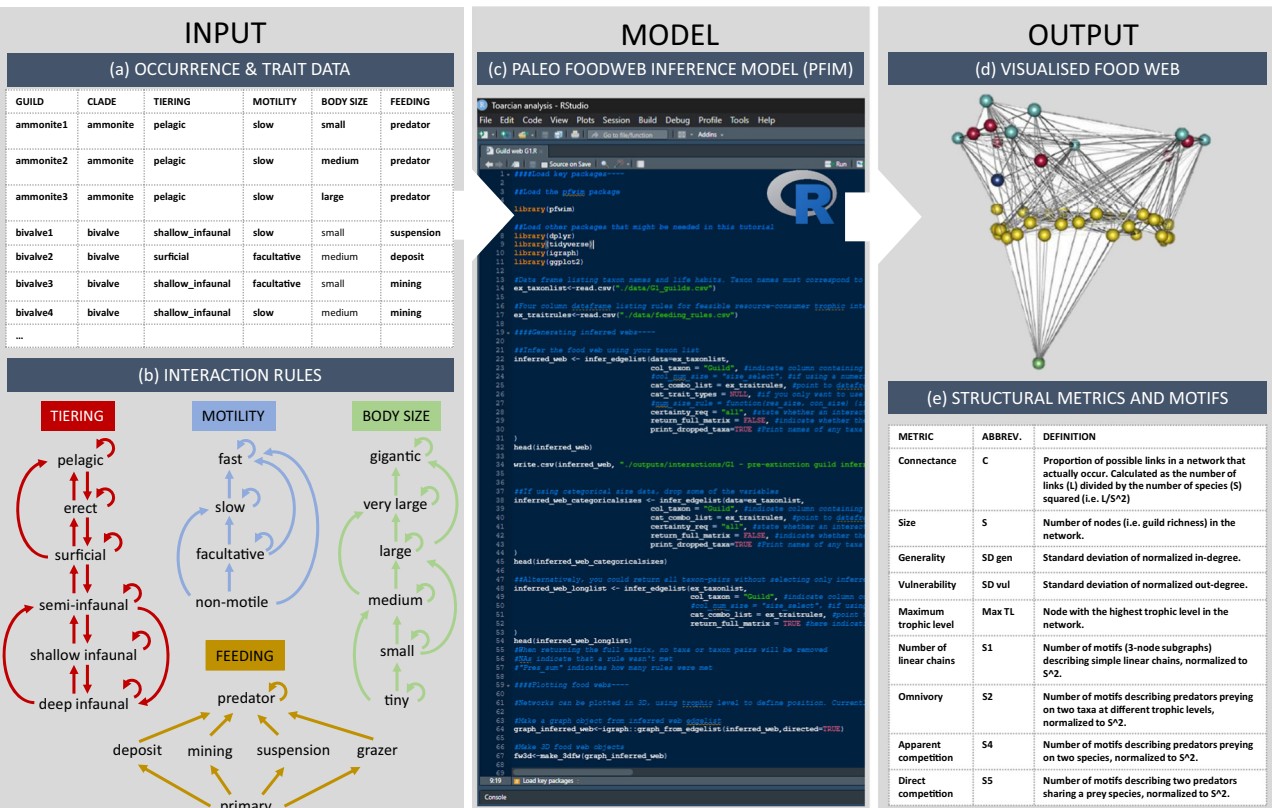

**Fig. 4 | Methodological schematic detailing the steps taken to reconstruct ancient food webs using the PFIM**[77]. INPUT: (**a**) community occurrence data and associated trait combinations for each taxon/guild occurrence; (**b**) trait-based feeding rules that parameterise PFIM for reconstructing empirical food webs across all intervals. MODEL: (**c**) explanation of how the model works and which methods of trophic network reconstruction were used in this study. OUTPUT: (**d**) Details of visualised trophic network output and (**e**) network metrics and motifs quantified by the model that were used in this study.

mining, grazer), and size: gigantic (>500 mm), very large (>300–500 mm), large (>100–300 mm), medium (>50–100 mm), small (>10–50 mm), tiny (≤10 mm). Size categories are defined by the longest axis of the fossil, estimates of tracemaker size from trace fossils based on literature accounts, or by extrapolating the total length for belemnites from the preserved guard using established approaches[78,79]. For example, an ammonite with a shell diameter of 75 mm would be classed as a large, slow, pelagic predator (see Fig. 4a for more occurrences with trait information). A single node for primary producers was added to each food web to ensure that primary consumers were not considered as primary producers in the reconstructions. Feeding interactions were modelled between organisms using the Paleo Foodweb Inference Model (PFIM), an inferential trait-based model which assigns the possibility of encounter and consumption of prey items using rules defined by ecological foraging traits (i.e., motility, feeding, tiering, and size; Fig. 4b)[80]. PFIM is a new approach in a class of models that have used functional traits to define ecological guilds and that are commonly employed in the fossil record. These functional traits have been used, for instance, to assess changes in functional diversity through time, both across mass extinction events[6,50,81] and to compare ancient and modern systems[82]. Within the model, feeding interactions were only realised if all feeding rules within each trait category were satisfied (see Fig. 4b) and the resultant webs represent trophic communities showing all possible feeding interactions, thus avoiding issues arising from ecological plasticity through time. Trophic guilds were defined by unique combinations of ecological and foraging traits (see Table S1 and Fig. 4a for a full list of trophic guilds and their defining characteristics) which correspond to

groups of organisms that have the same predation/prey rules dictating their interactions in the model and are thus akin to trophic species often used in the reconstruction of modern food webs[16,17,70]. Further palaeoecological data, which is used to inform the extinction cascade simulations, was also assigned to each trophic guild in the food web. This data included motility (fast, slow, facultative, non-motile), tiering (pelagic, epifaunal, infaunal), size (gigantic, very large, large, medium, small, tiny), and calcification (heavy, moderate, light).

### The Paleo Food Web Inference Model (PFIM)
The PFIM uses functional trait data- predictive of interactions in modern ecosystems and commonly available for fossil organisms- to reconstruct ancient food webs. Shaw et al.[80], tested the model by applying it to four modern marine ecosystems with empirical constrained food webs to directly compare PFIM-constructed networks to their empirical counterparts and found that (a) PFIM-inferred feasible food webs successfully predict ~70% of trophic interactions and (b) PFIM-inferred realised food webs accurately predict ~90% of interactions. PFIM is written in R, a commonly used coding language in ecology and palaeontology, and is simple to execute.

### Model input and interaction rules
PFIM uses occurrence lists of taxa/guilds (above) within a community together with ecological trait data (i.e., motility, tiering, feeding and body size) to define a plausible food web based on encounter feasibility and foraging biology (Fig. 4a). Life habit (motility, tiering and feeding) and size—together referred to as functional trait data- are key predictors of consumer-resource interactions and are easily defined

for metazoans. PFIM defines the feasibility of interaction between two taxa/guilds based on traits, creating a default set of interaction criteria (above) that determine whether a feeding interaction occurs in the model or not. All rules pertaining to all traits must be satisfied for an interaction to take place (Fig. 4b).

## How PFIM was used in this study

The PFIM can produce two different types of food web, (i) feasible webs; reflecting all potential interactions that might occur between taxa (a) over time and space according to the interaction rules (b); and (ii) realised webs; a flexible framework for generating hypothetical realised food webs with varying network characteristics, such as link-species distributions.

Feasible food webs are readily comparable with one another, permitting spatial and temporal evaluations of deep-time trophic trends, without any a priori assumptions about network structure whereas realised webs can be parameterised to match certain structural properties seen in modern communities.

In this study, we only use the feasible web approach, which reconstructs a single food web for each time interval consisting of all feasible feeding interactions- as determined by the functional traits of the organisms and the feeding rules presented in (a) and (b). The justification for this approach is three-fold. Firstly, we compare changes in trophic structure through an interval of geological time in order to capture the effects of a major extinction event on marine community structure and also to quantify the length and nature of the recovery interval. Thus, the use of the feasible web approach allows direct comparison of structural metrics between different time bins. Secondly, we want to capture all possible feeding interactions between all organisms in the food webs when simulating extinction cascades, as this accounts for organisms capabilities to switch between prey sources and for re-wiring to occur within the communities when taxa are lost. We feel this gives a better indication of community robustness to secondary extinction cascades than using the realised food web approach which will not capture all possible feeding interactions. Finally, we test for any permanent changes in community structure as a result to ongoing macroevolutionary changes in the Mesozoic Ocean. If we parameterise the PFIM to produce realised webs with modern-looking link-distributions, this will likely mask any changes in community structure that may have arisen as a result of ongoing macroevolutionary drivers such as the Mesozoic Marine Revolution.

## Quantifying community structure and function

PFIM output consists of visualisations of the food web network (Fig. 4d) and output of node-level and network-level metrics (Fig. 4e). Community network structural metrics of size (i.e., richness), connectance (C), maximum trophic level, generality (i.e., in-degree, or number of prey) and vulnerability (i.e., out-degree, or number of predators) as well as the network motifs S1 (i.e., number of linear chains), S2 (i.e., omnivory), S4 (i.e., apparent competition), and S5 (i.e., direct competition) were calculated to track changes in community structure and function across the extinction and through the recovery interval. Connectance, trophic level and generality/vulnerability are well established metrics linked to ecological function and and stability dynamics[64,65]. Motifs are a well-established tool to evaluate the distribution of interaction types and the presence or absence of indirect effects in communities[83].

## Simulating extinction cascades

Extinction cascades were simulated by subjecting guilds in the pre-extinction community to primary extinction scenarios based on ecological and trophic traits that correspond to known vulnerabilities linking the traits to hypothesised mass extinction drivers of anoxia, thermal stress and acidification. For each replicate, we catalogued the timing and identity of all primary extinctions and any secondary extinctions arising when a guild lost all of its resources. The extinction cascades were stopped when the diversity of the simulated post-extinction community reached 21 species and thus equalled that of the empirical post-extinction community.

We explored 13 different scenarios. Simulations were run with primary extinctions selected (i) randomly, (ii/iii) by body size (large to small/small to large), (iv/v) by tiering (infaunal to pelagic/pelagic to infaunal), (vi/vii) by motility (fast to non-motile/non-motile to fast), (viii/ix) calcification (heavy to light/light to heavy), (x/xi) generality (low to high/high to low), and (xii/xiii) vulnerability (low to high/high to low).

We implemented the modelling using the cheddar package in R[84,85] using the *RemoveNodes()* function with the 'cascade' method for secondary extinctions. We generated 50 replicates for each scenario by sampling among guilds from within each traits' levels in the sequence. For example, tiering has three levels (see above) and we randomised the primary extinction sequence of each guild within each of these levels.

Simulated post-extinction food webs were then compared to the empirical post-extinction community using three approaches. First, we compared nine structural metrics between the empirical post-extinction web and the simulated networks. Second, we compared the frequency of four motifs (S1: number of linear chains; S2: number of omnivory motifs; S4: number of apparent competition motifs; S5: number of direct competition motifs) between the empirical post-extinction web and the simulated networks. Third, we used a True Skill Statistic[74] (TSS/classification-misclassification table/confusion matrix: true positive, true negative, false positive, false negative) to compare the guild-node level similarities of position/identity between the empirical post-extinction web and the simulated networks. All calculations and analyses were in R version 4.2.2[86].

We combined the inference from all three of these comparisons to identify the most plausible set of primary extinction and associated secondary extinction scenarios (e.g., which trait sequence) that could deliver a community that most closely resembles the post-extinction community.

## Robustness analysis

Robustness was calculated for each of the four networks using a generalised method introduced in Jonsson et al.[66] where robustness, denoted as $R_x$, is defined as "the proportion of species that when, deleted primarily will result in x% of all species in the network/food web subsequently becoming extinct (as a result of primary, and secondary extinctions)". Here we estimate $R_x$ for x in 1:99%, creating 500 random replicates of primary extinction sequences and estimating the mean robustness and reporting the % of the original community remaining.

## Reporting summary

Further information on research design is available in the Nature Portfolio Reporting Summary linked to this article.

# Data availability

All the data used to construct the food webs are available at https://zenodo.org/records/11400588.

# Code availability

Reproducible code for the food web construction, extinction cascade simulations, structural analysis, motif analysis, robustness analysis and visualisations are available at https://zenodo.org/records/11400588. The full PFIM codebase for community reconstruction is currently under publication review[80].

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

## Acknowledgements

For the purposes of open access, the author has applied a Creative Commons Attribution (CC BY) licence (where permitted by UKRI, 'Open Government Licence' or 'Creative Commons Attribution No-derivatives (CC BY-ND) licence may be stated instead) to any Author Accepted Manuscript version arising'. This original pilot work for this study was funded by a Palaeontological Association Undergraduate Research Bursary (PA-UB01703) awarded to K.Z., C.T.S.L., J.W.A and A.M.D.; A.M.D. was funded by UKRI NERC grants NE/X015025/1 and NE/X012859/1. C.T.S.L. was funded by UKRI NERC grant NE/X015025/1. A.P.B. was funded by UKRI NERC grants NE/S001395/1, NE/T003502/1, NE/X015025/1 and NE/X012859/1.

## Author contributions

A.M.D. and A.P.B designed the study, analysed data, and wrote the manuscript. A.M.D., K.Z., C.T.S.L. and J.W.A. collected data. C.T.S.L. contributed to writing the manuscript. J.O.S. contributed to the analysis of data.

## Competing interests

The authors declare no competing interests.
