## [Transparent Peer Review file · Nature Communications]

Extinction cascades, community collapse, and recovery across a Mesozoic hyperthermal event

Corresponding Author: Dr Alexander Dunhill

This manuscript has been previously reviewed at another journal. This document only contains reviewer comments, rebuttal and decision letters for versions considered at Nature Communications.

Version 0:

Reviewer comments:

Reviewer #1

(Remarks to the Author)

The authors studied the ecosystem changes across the Early Toarcian Extinction Event, ~183 Ma. They found that there was a primary extinction selectivity based on ecological tier, with secondary extinctions concentrated amongst the secondary consumers that ate the primary consumers. This change is coupled to a change from a diverse, stable community, less diverse and stable community and that it took ~7 million years to recover from this extinction. The recovery ecosystem exhibited much higher levels of tiering, linking into the Mesozoic Marine Revolution, and modern marine ecosystem structure.

This work will be of significance to those working on the macroevolutionary dynamics of the Jurassic. For the work to be of interest to people outside the immediate field stronger links/more discussion needs to be made linking this study to the Marine Mesozoic Revolution and to modern ecosystem structure.

This is an interesting study, however there are fundamental flaws with the discussion regarding stability. The statement that stability decreases across the ETEE is not in evidence and may not be correct. Several of metrics associated with increased stability increase across the ETEE, so the opposite may be true. The study doesn't calculate any metrics of stability, and needs to do so, to be able to discuss stability. This lack of stability calculation, coupled with a lack of appreciation of what drives ecosystem stability (the authors appear to correlate it to connectance only, ignoring the interplay of connectance with species richness and interaction strength to drive stability) means that the interpretations that discuss ecosystem stability are not evidenced, and may be incorrect. Stability needs to be calculated prior to publication. See specific details below.

Detailed comments:

Abstract: For a general audience I think it would be useful to give approximate dates as well as the time intervals.

Line 47: Given that this paper is framed in terms of Court Jester and Red Queen, and abiotic vs biotic drivers of extinction dynamics I would suggest that the recent work by Spiridonov and Lovejoy Nature 2022 is worth citing/acknowledging since extinction dynamics form part of what is discussed in their study.

Line 54: Reference needed

Line 66: are tied?

Lines 99-107: More details about how the models are compared to the empirical data quantitatively would be good.

Line 153: Stable - you haven't defined how you have calculated this - if I understand correctly, you haven't calculated any stability metrics, it is just inferred from your food web structure. Stability is well established to correlate with complexity, so an increase in complexity (as stated in line 155) would suggest that stability increases across the ETEE, not decrease (which is what you infer).

Similarly, the metrics such as omnivory (lines 178-180) are correlated with increased stability (Neutel et al. Nature, 2007),

not a decrease, as you seem to imply.

Lines 187 - 191: Is this calculated or inferred? since things like omnivory have been shown to increase stability (e.g. Emmerson and Yearsley 2004, Neutel et al. 2007), it seems like this is suggested/inferred rather than calculated and could not be correct for this system.

Lines 194-197 - this behaviour has been shown to be stabilising.

Line 202: The patterning of energy flows within food webs is crucial for the maintenance of ecosystem stability (e.g. Rooney and McCann, TREE, 2012), with linear chains having the opposite effect of omnivory. Given that energy flows are not calculated within the study, it is not clear at all whether line 204 "removal of a few well-connected guilds could lead to wholesale ecosystem collapse" would occur or not.

Line 206: vulnerability is defined as number of predators. As such, it isn't clear here whether the authors are referring to this metric, or more general vulnerability, as linked to stability.

Line 162: What previous work? Please expand on what was known before your work, and what novel insights are provided by this study.

Line 190-211: Omnivory can increase stability (Emmerson and Yearsley, Proc. Roy. Soc. B, 2004)

Line 248: Complexity is normally described as a function of connectance, species richness and species interaction strengths (May 1972) so it isn't really appropriate to talk about complexity just in terms of connectance, or indeed link it to stability.

Line 313-315: Stability has not been calculated, and indeed may not decrease across the ETEE. As such, the following statement is not in evidence: "The extinction event is represented by a switch from a diverse, stable community where each key ecological function is performed by a number of guilds to a less diverse, more densely connected and less stable community of generalist "disaster taxa"."

Reviewer #2

(Remarks to the Author)

Although ecosystem-based approaches have long been recognised as essential for answering important ecological questions, paleoecology has lagged behind these advances. The manuscript by A. Dunhill et al. presents an important contribution to this end: they explore in depth the role of species interactions and community structure, beyond composition, in the Early Toarcian mass extinction and subsequent recovery. The most important results of this study are:

1) The mass extinction resulted in a shift from a stable, diverse community with many specialists, before the event, to a less stable community comprising mostly generalists, after recovery. The result that the new community was more connected than the original one is particularly significant.

2) Ecosystem functioning recovered faster than community structure.

3) Full ecosystem recover to pre-extinction levels took quite long, ~ 7 million years.

I find the author's results well supported by the evidence and backed by available literature. Regarding the details provided in the methods, I would like to see more reference to the stability of the assigned trait values over time.

These results presented in this manuscript have many implications for ecosystem resilience that go beyond the Early Toarcian, and are relevant to modern climate-induced extinctions. I look forward to seeing this paper published in Nature Communications.

Reviewer #3

(Remarks to the Author)

General Comments

The authors use a large dataset of species assemblages across a known marine extinction event to document the disassembly and re-assembly of a marine community in response to large-scale abiotic changes. Importantly, they argue that the extinction and recovery must be understood in terms of the ecological dynamics between species - that primary extinctions in response to abiotic changes resulted in secondary extinctions, and through this pathway the community changed over time. They find that a particular set of primary extinctions, defined by species traits and the sensitivities of those traits with respect to the changing environment, resulted in simulations that bore a strong resemblance to the extinction and eventual recovery of the empirical system.

I really enjoyed this paper - there are a lot of important messages, in particular that extinction resulted in a less complex, densely connected community with low redundancy, that eventually recovered to one that attains structural similarities with contemporary marine communities. The narrative is fairly clear in their conclusions, however the authors are not very clear in describing how they reconstruct food webs generally speaking. In part, this is because the methods they are using are in review elsewhere, however I think that they need to provide enough detail for the reader to understand how the reconstructions are tackled given that it is these reconstructions that form the entire basis of the paper's results.

I also felt that, here and there, the results are sold a bit strongly. The simulation results lead to networks that have similar identities to empirical post-extinction nets, and this is good support that the mechanisms they discuss and simulate are reasonable potential mechanisms to have operated in nature. But it is not direct evidence that these mechanisms were at work - some minor tweaks to the writing could address this issue.

In addition, there are a lot of interesting observations that the authors make with regard to the impact of large-scale extinction events on the structure of communities - many of these observations have also been made with respect to extinction events in other systems. That similar outcomes may characterize extinction events in very different systems (with teams using very different methods), certainly builds the argument that these outcomes may be general in nature, and it would make sense to connect this work to some of those.

Altogether, I found the paper interesting, of general importance (though this could be sold a bit better!), and what was explained was well explained.

Specific Comments

Some unstated assumptions - I think that one big one (unless I am mistaken) is that the same ecological interactions among guilds is assumed pre-and post-extinction... in other words, the same groups of species are assumed to interact in the same way, despite large changes in the environment and community. So while the community composition will change and result in different structures, the structures are inferred from the same set of rules... at least I am inferring this, as there is very little information provided with respect to how the food webs are assembled (the subject of a manuscript that is in press elsewhere). If this is correctly stated, I think the authors need to confront it right out of the gate and make a good argument for how their methods deal with ecological plasticity in how groups interact with each other.

Throughout - pay close attention to grammar... there are multiple occasions where singular vs. plural are mixed up a bit.

L52-54: I'm not sure if this is correctly phrased. Does the Red Queen hypothesis explicitly state this (this is how it reads), or are the conclusions or outcomes of this general theory pointing towards a community analysis highlighting important functional relationships operating within extinction/recovery? I'd guess the authors agree that it is the latter - if so, consider revising this statement.

L55: There seems to be a lot of evidence in contemporary systems that abiotic stress can have a large effect on extinction rates. This isn't to diminish the effects of secondary extinction, but this phrase seems a bit overly broad and not well-supported in our understanding of the contemporary world.

L111: I'd suggest defining infauna for readers outside the field

L110: I think it would be helpful to expand what you mean by 'tiering'... and given that this provides the closest approximation of the observed extinction/recovery sequence, consider providing a description of what would be occurring in these habitats by way of the 'tiering' scenario. (Similar to what you do on L130-144)

L118-120: I'm a little confused with respect to 'comparing empirical and simulated post-extinction structure' alongside species identity. Because the structural relationships connecting species are a function of their identity, wouldn't extinction simulations that result in communities with the same IDs (empirical vs. simulated) also have to have the same structures between them? I.e. because we have already established A->B, and A and B are simulated to survive, matching observations of A and B, we have already established A->B so that will be expected to match. So if I am understanding that correctly, the structural metrics aren't independent of identities. That would mean that communities with matching taxa (empirical vs. simulated) are expected to have similar structures, so the latter isn't really evidence that the simulations got it right. Or am I misunderstanding something?

L126-130: Wow - very interesting!

L134: 'These' scenarios also 'generate'

L153-157: Consider linking these observations with other works that have explored the structural consequences of extinction, where some similarities appear to emerge. For example, Pires et al (Quart Sci Rev; 2021) show that extinction cascades lead to structurally simpler/densely connected food web; Yeakel et al. (PNAS; 2014) found that the loss of structural redundancy followed mammalian extinctions throughout the Holocene; Dunne et al. (ProcRoySoc; 2014)

documented the recovery of Eocene food webs 18 Myrs after the KPg and the relatively quick re-attainment of largely modern structural characteristics; Fricke et al. (Science; 2022) show a loss in complexity & redundancy over the past 130 Kyrs in terrestrial systems. Despite these being very different systems, there do appear to be some interesting and aligning patterns, which might be particularly important given the differences in techniques being used. This would also serve to connect this work more generally to a larger body of literature.

L166: I'd be a bit careful here... your simulations suggest that the primary/secondary extinction sequence that you explore can give rise to observed patterns in empirical systems... but this is not direct evidence that this was the case. So I think the most you can say from this particular exploration is that the mechanisms explored here are clearly plausible candidates for what actually happened.

L369: I suppose the details are in the Shaw et al. paper that is in revision (so I can't evaluate here), but I think a little more information is required with regard to food web production. Is a single food web produced for each time interval, or a food web ensemble based on link probabilities, or something similar? While the full approach of the PFIM is the subject of another contribution, I think we still need enough details provided here for this contribution to stand on its own... with the current iteration, I don't have a clear sense of that.

L378-380: Why were these structural descriptors chosen over others? To what degree are the overall results sensitive to the selection of these metrics? Some description of why these metrics are particularly important I think, as well as an assessment of whether the results are more or less sensitive to a subset of them...

L390: the 21-species end-condition seems a bit arbitrary to me. 21 species should be considered a minimum, as there certainly could be other species that survive in low abundance that aren't documented. This probably doesn't matter too much, as those species would then be more likely to go extinct later on, but variability in who is actually present (perhaps in low abundances) during the recovery period could have an impact on interpretation of the validity of different cascade scenarios (perhaps). Do you think that potential variability in 'who is actually extinct in a given web' could alter the results?

Version 1:

Reviewer comments:

Reviewer #1

(Remarks to the Author)

All the points raised in my previous review have been well addressed. I appreciate the substantial amount of work done producing measures of robustness as a proxy for stability, which well addressed my previous concerns.

Reviewer #2

(Remarks to the Author)

I reviewed the original manuscript as well, and I am very satisfied with all the changes and additions the authors made.

Reviewer #3

(Remarks to the Author)

General comments

The authors do a nice job in addressing the points raised in the previous reviews. In this revision, there is still an elephant in the room, which is the PFIM used to establish the interaction network. I really don't have an issue with the PFIM approach itself - it appears to be a useful method to reconstruct trophic structure from the linked bioRxiv paper, at least as long as one assumes there is a direct correlation between the trait hierarchy and feeding (and here I don't think that's an issue). The main issue that I have (see below) is that the approach is not described well, and, to my mind, will be very confusing to readers unless they have also read the PFIM paper. How is figure 4 used in building the webs, specifically? As I mention below, this is really only tackled in a single sentence (or did I miss something)? Moreover, even with looking at the main PFIM paper, I'm a bit confused... are single webs being constructed for each period, or is an ensemble constructed? The text implies single webs, though the PFIM appears to support the creation of multiple hypothetical webs (an ensemble) based on a few assumptions about the link distribution function and number of realized links. Those details, and the assumptions used, need to be contained within this paper (in my opinion), even while the detailed justification for the PFIM approach is elsewhere. Related, the code and data should be archived using a tool such as Zenodo rather than a dynamic repository such as GitHub... it's great to link a GitHub repo, but the archived version should be included as well. Along those lines, and because the PFIM underlies everything else in this paper, I do not think excluding those methods and saying they would be attached to GitHub at a later date is appropriate.

All of that said, I do think the questions tackled and results described point to some interesting and exciting conclusions... this is a cool paper, well written, well thought out, and of potential interest to both contemporary and paleo ecologists.

Currently, the strength of the paper is diminished a bit by not clearly describing the central methodology the authors use to get to those exciting places. Without diminishing at all those results, I do think the authors need to be more rigorous in describing the tools and methods they use to reconstruct their food webs.

Specific comments

L80 - the edited sentence is phrased a bit awkward - consider revising, but how the authors define stability is more clear

L114 - this revision is more clear with respect to some of the terminology used. One thing that I think is still missing a bit is a general description of the community along these time periods. What is the 'ecological theater' - is this an intertidal community, shallow ocean basin? What are the dominant species before and after the extinction? I think that building a bit of a picture of the system would help us to interpret the author's findings. Of course this is just a suggestion. I do think that the parenthetical statements are a bit overdone where at times one parenthetical statement follows another (see L116). While these appear to be responses to the reviews, I would think they could be worked into the text a bit more seamlessly.

—

Methods

L372 - would be helpful to provide estimated times along side the interval stages - not many folks know when the upper Toarcian ends! (myself included) - this is done in the abstract, but it would be useful in the paper as well.

L376 - These assemblages - how much time do each of these bins represent? From figure 1 the depth is represented, but there isn't a clear way that I seem to be able to extract the temporal size of these bins. If there is good depth information, does changing these bins have any impact on the reconstructed structure? In other words, if you sliced the Early recovery period in half, does the first half look very different from the second half, and could the combined halves be representing a community that was never realized? In other words, what is the impact of the assumed time integration?

L388 - again knowing what general types of species we are dealing with would make this ecospace model and the traits included mean a bit more.

L402-414 - I've taken a look at the PFIM approach, and it seems like a really useful one for taking higher-dimensional trait information for an assemblage and putting together a hypothesized structure - particularly in food webs where trait information can be clearly linked to interactions. What's less clear here is whether or not these reconstructed food webs are single realizations or an ensemble of potential realizations. The revision on L419 makes it sound like there is a single representative structure being put together for each time period. This - again - is largely due to the authors not detailing how their PFIM is being used to put together the networks across time periods. Figure 4 is really useful for me to see because I've read their PFIM paper. I don't think I'd have any understanding of what that figure represents based only on the text in this paper, where we are really only guided by the single sentence: "Within the model, feeding interactions were only realised if all feeding rules within each trait category were satisfied (see Fig. 4)". Or did I miss something? Because this approach is the absolute *heart* of this paper, I truly feel that the authors need to find a way to better convey how the approach in Figure 4 connects to establishing the links within the food webs in Figure 1. There are also some potential concerns that I can see with the PFIM approach (if I understand it correctly). That is, the strict realization of feeding links based on fulfilling *all* feeding rule criteria assumes that these interactions represent 'potential' links in the system, allowed by our understanding of how traits interoperate. That does not mean these links would have all been realized.

In the PFIM paper, the authors state: "PFIM generates a series of replicate hypothetical realized webs using a link distribution function to reduce the feasible links assigned to each node to match a hypothetical distribution and number of realized links."... is this being used here? As mentioned above, the text makes it sound like only a single realization is being used in each time period, but the PFIM approach would appear to allow for some statistical ensemble to represent a food web reconstruction. And going back to the PFIM statement, how is this hypothetical distribution/number of realized links chosen, and how sensitive are the results to this choice? None of my questions should indicate that the paper is substantially flawed, but I don't think my confusion on these points is unwarranted. PFIM is a new approach, and I think a paper that is premised on its use should clearly define how it is being used to the extent that the results could be replicated by another independent group...

L483 - In my opinion, this does not follow best practices (I've been corrected on this recently myself). The full approach should be included for this publication (not promised at a future date), and should not be a dynamic GitHub repository (which is not archived and can change on the whim of a contributor). The code/data/etc for this paper should be archived with a tool such as zenodo (or similar), which works within Github to archive the copy of the code that directly relates to the final published version of this paper... that way the version of the code that is linked to a contribution is set in stone, while still linked to the dynamic repository as updates are introduced. Moreover, I think this paper in particular leans so heavily on the PFIM approach, it really should not be separated from the rest of the analyses as it appears to be from L479-481.

L470 - how are secondary extinctions defined? Maybe I missed this - do species have to lose *all* links to go secondarily extinct? I'd imagine a species losing its predators but not its prey would remain? Could a generalist lose 4/5 of its prey and remain?

—

L254 - "...might *be* expected..." perhaps?

L295 - MMR should be defined earlier (line 230)

L313 - I'd specify that this is 'structural robustness', not to be confused with dynamic robustness... as far as primary/secondary extinctions go, I find the definition on L317 a little confusing in how primary vs. secondary extinctions are counted towards that '50'... in other words, would 10% species suffering primary extinction leading to 40% species going secondarily extinct count the same as 40% species suffering primary extinction leading to 10% species going secondarily extinct? Those both erode 50% of the system (R50), but would imply a very different type of structural robustness. I understand that this is an approach developed elsewhere, but do those 2 scenarios being ranked the same, but implying something different about robustness, impact our interpretations of extinction scenarios?

Reviewer #1 (Remarks to the Author):

The authors studied the ecosystem changes across the Early Toarcian Extinction Event, ~183 Ma. They found that there was a primary extinction selectivity based on ecological tier, with secondary extinctions concentrated amongst the secondary consumers that ate the primary consumers. This change is coupled to a change from a diverse, stable community, less diverse and stable community and that it took ~7 million years to recover from this extinction. The recovery ecosystem exhibited much higher levels of tiering, linking into the Mesozoic Marine Revolution, and modern marine ecosystem structure.

Thank you for this accurate assessment of our work.

This work will be of significance to those working on the macroevolutionary dynamics of the Jurassic. For the work to be of interest to people outside the immediate field stronger links/more discussion needs to be made linking this study to the Marine Mesozoic Revolution and to modern ecosystem structure.

Thank you for this assessment. We have expanded, on Lines 295-305 and 331-338, the discussion and relevance to the MMR and modern ecosystems and introduced this more formally early in the MS in Lines 225-233.

This is an interesting study, however there are fundamental flaws with the discussion regarding stability. The statement that stability decreases across the ETEE is not in evidence and may not be correct. Several of metrics associated with increased stability increase across the ETEE, so the opposite may be true. The study doesn't calculate any metrics of stability, and needs to do so, to be able to discuss stability. This lack of stability calculation, coupled with a lack of appreciation of what drives ecosystem stability (the authors appear to correlate it to connectance only, ignoring the interplay of connectance with species richness and interaction strength to drive stability) means that the interpretations that discuss ecosystem stability are not evidenced, and may be incorrect. Stability needs to be calculated prior to publication. See specific details below.

Thank you for the detailed reading and sharing insights about stability here and in specific sections below. You are correct that we have not specifically calculated any stability metrics from the networks. We address the comment above and detailed ones below in the following way. As you point out, knowing C, omnivory and motif details is likely insufficient to establish patterns of stability without reference to interaction strength magnitudes and distributions. This is because the core theory from May, derivatives from Allesina and Tang and from deRuiter and Neutel are all based on defining complexity and stability in the framework of S, C and interaction strengths. Furthermore, specific linking to motifs is challenging as the trophic chain ideas alone are too loosely tied to Rooney and McCann theory about fast and slow energy channels linked to flux and the distribution of interaction strengths.

Therefore, instead of attempting to link these metrics to stability, we implemented a network robustness analysis to more formally assess an established indicator of 'stability', though we restrict the MS to the term robustness. Based around the R50 metric from secondary extinction work in ecology (R50 is = the proportion of species that when, deleted primarily will result in 50% of all species in the network/food web subsequently becoming extinct (as a result of primary, and secondary extinctions)), we follow Jonsson et al's generalisation of this across a gradient of primary

extinction values (e.g. 1-99%). This analysis of each network provides a uniform and established approach to compare an established metric that links structure to a measure of stability common now in modern ecology.

Because this new analysis spans all four networks, we place this as the final section in our work (lines 307-339) and introduce a new figure 3 to the manuscript).

Detailed comments:

Abstract: For a general audience I think it would be useful to give approximate dates as well as the time intervals. *We have added this information as requested (line 25).*

Line 47: Given that this paper is framed in terms of Court Jester and Red Queen, and abiotic vs biotic drivers of extinction dynamics I would suggest that the recent work by Spiridonov and Lovejoy Nature 2022 is worth citing/acknowledging since extinction dynamics form part of what is discussed in their study. *Reference added (line 48).*

Line 54: Reference needed *We thank the referee for this request, and have provided Huang et als' recent work as a reference (line 54).*

Line 66: are tied? *Corrected (line 66).*

Lines 99-107: More details about how the models are compared to the empirical data quantitatively would be good. *We have modified this section to be more clear about the 'who died and why' statement, which we think is causing confusion. This now reads: "comparing the predicted loss of guilds from our secondary extinction modelling to existing empirical knowledge of extinction selectivity over these time periods." (lines 106-109). We have also modified the next statement, which emphasises the separate analysis of structure and function (see lines 108-111). Full details are provided in the Methods section.*

Line 153: Stable - you haven't defined how you have calculated this - if I understand correctly, you haven't calculated any stability metrics, it is just inferred from your food web structure. Stability is well established to correlate with complexity, so an increase in complexity (as stated in line 155) would suggest that stability increases across the ETEE, not decrease (which is what you infer). *See above – removed and replaced with formal robustness analysis*

Similarly, the metrics such as omnivory (lines 178-180) are correlated with increased stability (Neutel et al. Nature, 2007), not a decrease, as you seem to imply. *See above – removed and replaced with formal robustness analysis*

Lines 187 - 191: Is this calculated or inferred? since things like omnivory have been shown to increase stability (e.g. Emmerson and Yearsley 2004, Neutel et al. 2007), it seems like this is suggested/inferred rather than calculated and could not be correct for this system. *See above – removed and replaced with formal robustness analysis*

Lines 194-197 - this behaviour has been shown to be stabilising. *See above – removed and replaced with formal robustness analysis*

Line 202: The patterning of energy flows within food webs is crucial for the maintenance of ecosystem stability (e.g. Rooney and McCann, TREE, 2012), with linear chains having the opposite effect of omnivory. Given that energy flows are not calculated within the study, it is not clear at all whether line 204 "removal of a few well-connected guilds could lead to wholesale ecosystem collapse" would occur or not.

See above – removed and replaced with formal robustness analysis

Line 206: vulnerability is defined as number of predators. As such, it isn't clear here whether the authors are referring to this metric, or more general vulnerability, as linked to stability.

See above – removed and replaced with formal robustness analysis

Line 162: What previous work? Please expand on what was known before your work, and what novel insights are provided by this study. *Apologies for the confusion - we have reworked this entire paragraph for clarity to show parallels between reference 39 and our simulation results (lines 166-168).*

Line 190-211: Omnivory can increase stability (Emmerson and Yearsley, Proc. Roy. Soc. B, 2004)

We have modified this section for clarity around how patterns of community structure and motif frequency would arise. The stability issues are dealt with elsewhere as noted above (lines 200-210).

Line 248: Complexity is normally described as a function of connectance, species richness and species interaction strengths (May 1972) so it isn't really appropriate to talk about complexity just in terms of connectance, or indeed link it to stability.

We have reverted to using connectance only here (line 247).

Line 313-315: Stability has not been calculated, and indeed may not decrease across the ETEE. As such, the following statement is not in evidence: "The extinction event is represented by a switch from a diverse, stable community where each key ecological function is performed by a number of guilds to a less diverse, more densely connected and less stable community of generalist "disaster taxa"."

See above – we have removed the word stability in this sentence and this section has been modified to include reference to the robustness analysis.

Reviewer #2 (Remarks to the Author):

Although ecosystem-based approaches have long been recognised as essential for answering important ecological questions, paleoecology has lagged behind these advances. The manuscript by A. Dunhill et al. presents an important contribution to this end: they explore in depth the role of species interactions and community structure, beyond composition, in the Early Toarcian mass extinction and subsequent recovery. The most important results of this study are:

- 1) The mass extinction resulted in a shift from a stable, diverse community with many specialists, before the event, to a less stable community comprising mostly generalists, after recovery. The result the the new community was more connected than the original one is particularly significant.
- 2) Ecosystem functioning recovered faster than community structure.
- 3) Full ecosystem recover to pre-extinction levels took quite long, ~ 7 million years.

I find the author's results well supported by the evidence and backed by available literature.

Regarding the details provided in the methods, I would like to see more reference to the stability of the assigned trait values over time. *We believe the referee is referring to whether our categorical*

functional traits used to construct feeding links and networks is applicable over the time periods we study in the fossil record. The PFIM method we use is a variation of such functional trait methods and such functional trait-based analyses are already commonly employed in the fossil record. These traits have been used, for instance, to assess changes in functional diversity through time, both across mass extinction events (Foster and Twitchett 2014; Dunhill et al. 2018) and to compare ancient and modern systems (Bush and Bambach 2011). The traits are categorical representations of habitat use and life-style that have persisted over geological time periods. We have added detail about this to the methods section relating to PFIM and clarified the text (lines 405-412).

These results presented in this manuscript have many implications for ecosystem resilience that go beyond the Early Toarcian, and are relevant to modern climate-induced extinctions. I look forward to seeing this paper published in Nature Communications.

We thank reviewer #2 for their positive feedback.

Reviewer #3 (Remarks to the Author):

General Comments

The authors use a large dataset of species assemblages across a known marine extinction event to document the disassembly and re-assembly of a marine community in response to large-scale abiotic changes. Importantly, they argue that the extinction and recovery must be understood in terms of the ecological dynamics between species - that primary extinctions in response to abiotic changes resulted in secondary extinctions, and through this pathway the community changed over time. They find that a particular set of primary extinctions, defined by species traits and the sensitivities of those traits with respect to the changing environment, resulted in simulations that bore a strong resemblance to the extinction and eventual recovery of the empirical system.

I really enjoyed this paper - there are a lot of important messages, in particular that extinction resulted in a less complex, densely connected community with low redundancy, that eventually recovered to one that attains structural similarities with contemporary marine communities. The narrative is fairly clear in their conclusions, however the authors are not very clear in describing how they reconstruct food webs generally speaking. In part, this is because the methods they are using are in review elsewhere, however I think that they need to provide enough detail for the reader to understand how the reconstructions are tackled given that it is these reconstructions that form the entire basis of the paper's results.

I also felt that, here and there, the results are sold a bit strongly. The simulation results lead to networks that have similar identities to empirical post-extinction nets, and this is good support that the mechanisms they discuss and simulate are reasonable potential mechanisms to have operated in nature. But it is not direct evidence that these mechanisms were at work - some minor tweaks to the writing could address this issue. *We have reviewed the strength of our language throughout the work to ensure we emphasise that the mechanisms could be at work, rather than that they are at work. The strength of our conclusions is now better aligned with the volume of evidence we present. In addition, we have added additional quantitative analyses of stability that add robustness to some of our conclusions.*

In addition, there are a lot of interesting observations that the authors make with regard to the

impact of large-scale extinction events on the structure of communities - many of these observations have also been made with respect to extinction events in other systems. That similar outcomes may characterize extinction events in very different systems (with teams using very different methods), certainly builds the argument that these outcomes may be general in nature, and it would make sense to connect this work to some of those. *We thank the referee for this comment and refer to their comment below with specific references provided.*

Altogether, I found the paper interesting, of general importance (though this could be sold a bit better!), and what was explained was well explained.

Specific Comments

Some unstated assumptions - I think that one big one (unless I am mistaken) is that the same ecological interactions among guilds is assumed pre-and post-extinction... in other words, the same groups of species are assumed to interact in the same way, despite large changes in the environment and community. So while the community composition will change and result in different structures, the structures are inferred from the same set of rules... at least I am inferring this, as there is very little information provided with respect to how the food webs are assembled (the subject of a manuscript that is in press elsewhere). If this is correctly stated, I think the authors need to confront it right out of the gate and make a good argument for how their methods deal with ecological plasticity in how groups interact with each other. *We thank the referee for this comment. We have enhanced the methods section detailing the PFIM approach to a) focus readers more carefully on the use of functional, categorical traits as a foundation for assigning potential feeding links and b) to be clear that composition drives the changes because the structures are inferred from the same set of rules throughout the time period (lines 405-412).*

Throughout - pay close attention to grammar... there are multiple occasions where singular vs. plural are mixed up a bit. *We have checked through for this.*

L52-54: I'm not sure if this is correctly phrased. Does the Red Queen hypothesis explicitly state this (this is how it reads), or are the conclusions or outcomes of this general theory pointing towards a community analysis highlighting important functional relationships operating within extinction/recovery? I'd guess the authors agree that it is the latter - if so, consider revising this statement. *We have clarified our second reference here to the RQH, now making more clear that we are talking about the extinction component: "Thus, modern ecological theory and the importance of extinction in the Red Queen hypothesis suggest that extinction and recovery dynamics to such hyperthermal events are likely to be most effectively evaluated via a community ecological framework." (lines 52-54).*

L55: There seems to be a lot of evidence in contemporary systems that abiotic stress can have a large effect on extinction rates. This isn't to diminish the effects of secondary extinction, but this phrase seems a bit overly broad and not well-supported in our understanding of the contemporary

world. *We agree with the reviewer that there is lots of evidence for abiotic primary extinctions. Our premise is that total extinction rates are 50-90% across major mass extinction events and that beyond 50% you are very likely to be estimating a mix of primary and secondary species extinction; systems will reach a point where primary extinctions result in secondary cascades. We believe we are clearly making the point here that secondary extinctions are only part of the process, but an important and often overlooked part of the process. Our paragraph provides numerous references justifying this.*

L111: I'd suggest defining infauna for readers outside the field. *Defined as "organisms living within the sediment" (lines 116).*

L110: I think it would be helpful to expand what you mean by 'tiering'... and given that this provides the closest approximation of the observed extinction/recovery sequence, consider providing a description of what would be occurring in these habitats by way of the 'tiering' scenario. (Similar to what you do on L130-144). *This is a great suggestion. We have expanded the explanation here and indicated also that there is additional discussion of the result linked to structural change below (lines 114-115, 118-120).*

L118-120: I'm a little confused with respect to 'comparing empirical and simulated post-extinction structure' alongside species identity. Because the structural relationships connecting species are a function of their identity, wouldn't extinction simulations that result in communities with the same IDs (empirical vs. simulated) also have to have the same structures between them? I.e. because we have already established A->B, and A and B are simulated to survive, matching observations of A and B, we have already established A->B so that will be expected to match. So if I am understanding that correctly, the structural metrics aren't independent of identities. That would mean that communities with matching taxa (empirical vs. simulated) are expected to have similar structures, so the latter isn't really evidence that the simulations got it right. Or am I misunderstanding something? *We believe there is a misunderstanding. There is a mapping of guild identity to structural characteristics on a gross scale such that the loss of all infaunal species, for example, would have a clear impact. However, our simulations are not about loss of the trait, but about guilds with the trait. We implement replicated, randomised sequences of loss that each lead to different outcomes. Our use of the TSS metric specifically allows us and readers to distinguish between identity effects and structural change. It happens to be that tiering delivers the strongest inference for both TSS and structural metrics, whereas others do not.*

L126-130: Wow - very interesting!

L134: 'These' scenarios also 'generate' *Corrected (line 142).*

L153-157: Consider linking these observations with other works that have explored the structural consequences of extinction, where some similarities appear to emerge. For example, Pires et al (Quart Sci Rev; 2021) show that extinction cascades lead to structurally simpler/densely connected food web; Yeakel et al. (PNAS; 2014) found that the loss of structural redundancy followed mammalian extinctions throughout the Holocene; Dunne et al. (ProcRoySoc; 2014) documented the recovery of Eocene food webs 18 Myrs after the KPg and the relatively quick re-attainment of largely

modern structural characteristics; Fricke et al. (Science; 2022) show a loss in complexity & redundancy over the past 130 Kyr in terrestrial systems. Despite these being very different systems, there do appear to be some interesting and aligning patterns, which might be particularly important given the differences in techniques being used. This would also serve to connect this work more generally to a larger body of literature. *We thank the reviewer for these suggestions, and we have now linked our general findings to these very relevant papers (line 165-168). We haven't included the Dunne et al 2014 paper in this section as they don't specifically test extinction effects or the nature of the recovery from the K-Pg, rather they form a hypothesis that the modern structure seen in the Eocene webs is representative of recovery following the K-Pg mass extinction event. A great hypothesis that we are currently attempting to test, but not as relevant to this study.*

L166: I'd be a bit careful here... your simulations suggest that the primary/secondary extinction sequence that you explore can give rise to observed patterns in empirical systems... but this is not direct evidence that this was the case. So I think the most you can say from this particular exploration is that the mechanisms explored here are clearly plausible candidates for what actually happened. *Added words to clarify that this is a likely scenario rather than a certain one (lines 176-180).*

L369: I suppose the details are in the Shaw et al. paper that is in revision (so I can't evaluate here), but I think a little more information is required with regard to food web production. Is a single food web produced for each time interval, or a food web ensemble based on link probabilities, or something similar? While the full approach of the PFIM is the subject of another contribution, I think we still need enough details provided here for this contribution to stand on its own... with the current iteration, I don't have a clear sense of that.

We've elaborated on how the PFIM works in the methods section (lines 405-414) and have now posted the PFIM manuscript on BioRxiv

(<https://www.biorxiv.org/content/10.1101/2024.01.30.578036v1>) so the detailed methodology is now easily accessible for reviewers and readers once this paper is published.

We've edited the wording to make it clear that a single food web was produced for each time interval (line 416).

L378-380: Why were these structural descriptors chosen over others? To what degree are the overall results sensitive to the selection of these metrics? Some description of why these metrics are particularly important I think, as well as an assessment of whether the results of more or less sensitive to a subset of them... *We have added justification for these:*

Connectance, trophic level and generality/vulnerability are well established metrics linked to ecological stability dynamics (Allesina and Tang, Rooney and McCann). Motifs are a well-established tool to evaluate the distribution of interaction types and the presence or absence of indirect effects in communities (Stouffer and Bascompte 2010) (lines 429-432).

L390: the 21-species end-condition seems a bit arbitrary to me. 21 species should be considered a minimum, as there certainly could be other species that survive in low abundance that aren't documented. This probably doesn't matter too much, as those species would then be more likely to go extinct later on, but variability in who is actually present (perhaps in low abundances) during the

recovery period could have an impact on interpretation of the validity of different cascade scenarios (perhaps). Do you think that potential variability in 'who is actually extinct in a given web' could alter the results? *A very insightful point from the reviewer for which we thank them. However, the preservation potential in the low-oxygen black shale environment is exceptional, so we very much doubt we are missing considerable portions of the post-extinction community that would invalidate the results of the modelling due to stopping the extinction cascades at the same taxonomic diversity as the empirical post-extinction community.*

REVIEWER COMMENTS

Reviewer #1 (Remarks to the Author):

All the points raised in my previous review have been well addressed. I appreciate the substantial amount of work done producing measures of robustness as a proxy for stability, which well addressed my previous concerns.

Reviewer #1 (Remarks on code availability):

results are appropriately reproducible and usable.

We thank reviewer 1 for their helpful and constructive reviews.

Reviewer #2 (Remarks to the Author):

I reviewed the original manuscript as well, and I am very satisfied with all the changes and additions the authors made.

We thank reviewer 2 for their helpful and constructive reviews.

Reviewer #3 (Remarks to the Author):

General comments

The authors do a nice job in addressing the points raised in the previous reviews. In this revision, there is still an elephant in the room, which is the PFIM used to establish the interaction network. I really don't have an issue with the PFIM approach itself - it appears to be a useful method to reconstruct trophic structure from the linked bioarxiv paper, at least as long as one assumes there is a direct correlation between the trait hierarchy and feeding (and here I don't think that's an issue). The main issue that I have (see below) is that the approach is not described well, and, to my mind, will be very confusing to readers unless they have also read the PFIM paper. How is figure 4 used in building the webs, specifically? As I mention below, this is really only tackled in a single sentence (or did I miss something)? Moreover, even with looking at the main PFIM paper, I'm a bit confused... are single webs being constructed for each period, or is an ensemble constructed? The text implies single webs, though the PFIM appears to support the creation of multiple hypothetical webs (an ensemble) based on a few assumptions about the link distribution function and number of realized links. Those details, and the assumptions used, need to be contained within this paper (in my opinion), even while the detailed justification for the PFIM approach is elsewhere. Related, the code and data should be archived using a tool such as Zenodo rather than a dynamic repository such as GitHub... it's great to link a GitHub repo, but the archived version should be included as well. Along those lines, and because the PFIM underlies everything else in this paper, I do not think excluding those methods and saying they would be attached to GitHub at a later date is appropriate.

We agree with the reviewer that we did not adequately explain every step used in the PFIM reconstructions. We have addressed this by adding further detail to the methods section and by replacing the old Figure 4 with a more comprehensive methods box figure that outlines every step

from data input to model output. We have detailed that we have indeed only used a single metaweb reconstruction for each time interval and we explain why we have chosen to do this.

All of that said, I do think the questions tackled and results described point to some interesting and exciting conclusions... this is a cool paper, well written, well thought out, and of potential interest to both contemporary and paleo ecologists. Currently, the strength of the paper is diminished a bit by not clearly describing the central methodology the authors use to get to those exciting places. Without diminishing at all those results, I do think the authors need to be more rigorous in describing the tools and methods they use to reconstruct their food webs.

We thank reviewer 3 for their insightful reviews that have greatly improved this manuscript.

Specific comments

L80 - the edited sentence is phrased a bit awkward - consider revising, but how the authors define stability is more clear

Added brackets around “measured as robustness to extinction” so sentence reads better.

L114 - this revision is more clear with respect to some of the terminology used. One thing that I think is still missing a bit is a general description of the community along these time periods. What is the ‘ecological theater’ - is this an intertidal community, shallow ocean basin? What are the dominant species before and after the extinction? I think that building a bit of a picture of the system would help us to interpret the author’s findings. Of course this is just a suggestion. I do think that the parenthetical statements are a bit overdone where at times one parenthetical statement follows another (see L116). While these appear to be responses to the reviews, I would think they could be worked into the text a bit more seamlessly.

This is a good suggestion and we have added detail to the intro section (lines 94-95) to show what types of invertebrates make up these communities where we also state that this is a shallow continental shelf community (line 91). We have dialled down the parenthetical statements in the opening section of the “Who died and why?” results section.

—

Methods

L372 - would be helpful to provide estimated times along side the interval stages - not many folks know when the upper Toarcian ends! (myself included) - this is done in the abstract, but it would be useful in the paper as well.

Good point – rough dates added as it is difficult to date this section absolutely (lines 373-374).

L376 - These assemblages - how much time do each of these bins represent? From figure 1 the depth is represented, but there isn’t a clear way that I seem to be able to extract the temporal size of these bins. If there is good depth information, does changing these bins have any impact on the reconstructed structure? In other words, if you sliced the Early recovery period in half, does the first half look very different from the second half, and could the combined halves be representing a

community that was never realized? In other words, what is the impact of the assumed time integration?

The communities representing each of these time intervals would not greatly change if you sliced them up into smaller sections, albeit with rarer taxa being lost from some of the smaller sections. It is difficult to put dates to these sections as they have not been radiometrically dated and are based on biostratigraphy. That said, there will be some time averaging effects, however our binning is consistent with that of Atkinson et al. (2023) and the bins are defined by observable changes in taxonomic diversity and ecological structure.

L388 - again knowing what general types of species we are dealing with would make this ecospace model and the traits included mean a bit more.

To help add some extra clarity, we have added an example of how an organism would be classified using the defined ecological traits (lines 400-401). This is also covered in the methods box (new figure 4) which details a subset list of one of the occurrence lists with traits.

L402-414 - I've taken a look at the PFIM approach, and it seems like a really useful one for taking higher-dimensional trait information for an assemblage and putting together a hypothesized structure - particularly in food webs where trait information can be clearly linked to interactions. What's less clear here is whether or not these reconstructed food webs are single realizations or an ensemble of potential realizations. The revision on L419 makes it sound like there is a single representative structure being put together for each time period. This - again - is largely due to the authors not detailing how their PFIM is being used to put together the networks across time periods. Figure 4 is really useful for me to see because I've read their PFIM paper. I don't think I'd have any understanding of what that figure represents based only on the text in this paper, where we are really only guided by the single sentence: "Within the model, feeding interactions were only realised if all feeding rules within each trait category were satisfied (see Fig. 4)". Or did I miss something? Because this approach is the absolute *heart* of this paper, I truly feel that the authors need to find a way to better convey how the approach in Figure 4 connects to establishing the links within the food webs in Figure 1. There are also some potential concerns that I can see with the PFIM approach (if I understand it correctly). That is, the strict realization of feeding links based on fulfilling *all* feeding rule criteria assumes that these interactions represent 'potential' links in the system, allowed by our understanding of how traits interoperate. That does not mean these links would have all been realized.

Yes, the reviewer is correct that we opted to use a single metaweb for each time bin. We have now detailed the reasons for this choice in the new version of Figure 4, which is a detailed methods box for how we use the PFIM and how it ultimately models community structure based on fossil occurrences and traits. The text is reproduced below for the benefit of the reviewer and editor:

"In this study we only use the feasible web approach, which reconstructs a single food web for each time interval consisting of all feasible feeding interactions- as determined by the functional traits of the organisms and the feeding rules presented in (a) and (b). The justification for this approach is three-fold. Firstly, we compare changes in trophic structure through a interval of geological time in order to capture the effects of a major extinction event on marine community

structure and also to quantify the length and nature of the recovery interval. Thus, the use of the feasible web approach allows direct comparison of structural metrics between different time bins.

Secondly, we want to capture all possible feeding interactions between all organisms in the food webs when simulating extinction cascades, as this accounts for organisms capabilities to switch between prey sources and for re-wiring to occur within the communities when taxa are lost. We feel this gives a better indication of community robustness to secondary extinction cascades than using the realised food web approach which will not capture all possible feeding interactions.

Finally, we test for any permanent changes in community structure as a result to ongoing macroevolutionary changes in the Mesozoic ocean. If we parameterise the PFIM to produce realised webs with modern-looking link distributions, this will likely mask any changes in community structure that may have arisen as a result of ongoing macroevolutionary drivers such as the Mesozoic Marine Revolution.”

The methods box also puts the old figure 4 into context i.e. the whole process (included the feeding rules schematic) is now detailed from the data input stages, through the modelling stage, to the output stage.

RE the feasible links approach, again, we feel that it is more appropriate to include all feasible links, particularly when testing for secondary extinction as this allows the potential for re-wiring of the food webs if preferred prey are lost.

In the PFIM paper, the authors state: “PFIM generates a series of replicate hypothetical realized webs using a link distribution function to reduce the feasible links assigned to each node to match a hypothetical distribution and number of realized links.”... is this being used here? As mentioned above, the text makes it sound like only a single realization is being used in each time period, but the PFIM approach would appear to allow for some statistical ensemble to represent a food web reconstruction. And going back to the PFIM statement, how is this hypothetical distribution/number of realized links chosen, and how sensitive are the results to this choice? None of my questions should indicate that the paper is substantially flawed, but I don't think my confusion on these points is unwarranted. PFIM is a new approach, and I think a paper that is premised on its use should clearly define how it is being used to the extent that the results could be replicated by another independent group...

The response above answers this concern as well.

L483 - In my opinion, this does not follow best practices (I've been corrected on this recently myself). The full approach should be included for this publication (not promised at a future date), and should not be a dynamic GitHub repository (which is not archived and can change on the whim of a contributor). The code/data/etc for this paper should be archived with a tool such as zenodo (or similar), which works within Github to archive the copy of the code that directly relates to the final published version of this paper... that way the version of the code that is linked to a contribution is set in stone, while still linked to the dynamic repository as updates are introduced. Moreover, I think this paper in particular leans so heavily on the PFIM approach, it really should not be separated from the rest of the analyses as it appears to be from L479-481.

We thank the referee for this. The Zenodo details are now provided as the canonical link to the information (see lines 484-487 in revised document). We have made clear that all the steps in the

analysis can be completed with the code in this repository (including the PFIM functions and operational code for network generation).

L470 - how are secondary extinctions defined? Maybe I missed this - do species have to lose **all** links to go secondarily extinct? I'd imagine a species losing its predators but not its prey would remain? Could a generalist lose 4/5 of its prey and remain?

The referee is correct. The history of modelling secondary extinctions in ecology using topological (non-dynamic models) defines secondary extinction as occurring when a consumer loses all of its prey. Our references to these models/approaches in the MS reflect this assumption. We have added the definition to the start of the Results and Discussion section too.

—

L254 - "...might **be** expected..." perhaps?

Corrected.

L295 - MMR should be defined earlier (line 230)

Corrected.

L313 - I'd specify that this is 'structural robustness', not to be confused with dynamic robustness... as far as primary/secondary extinctions go, I find the definition on L317 a little confusing in how primary vs. secondary extinctions are counted towards that '50'... in other words, would 10% species suffering primary extinction leading to 40% species going secondarily extinct count the same as 40% species suffering primary extinction leading to 10% species going secondarily extinct? Those both erode 50% of the system (R_{50}), but would imply a very different type of structural robustness. I understand that this is an approach developed elsewhere, but do those 2 scenarios being ranked the same, but implying something different about robustness, impact our interpretations of extinction scenarios?

We thank the referee for the suggestion about structural robustness and have added this wording to the MS lines 391 - 321.

" R_{50} , the proportion of species that need to be made extinct via primary extinction that leads to a 50% loss of all species in the network/food web as a result of primary, and secondary extinctions (see ^{66,67})."

So while both of the referee's scenarios lead to the same outcome (the 50 part of R_{50}), the value of R_{50} is different (10 versus 40) and this, we believe, reflects the distinction the referee is searching for. The 50 is the total target loss, the R_{50} value is the indicator of different robustnesses.